# Identification of *CNGCs* in *Glycine max* and Screening of Related Resistance Genes after *Fusarium solani* Infection

**DOI:** 10.3390/biology12030439

**Published:** 2023-03-12

**Authors:** Yuxing Cui, Jingxuan Wang, Yingxue Bai, Liping Ban, Junda Ren, Qiaoxia Shang, Weiyu Li

**Affiliations:** 1College of Plant Science and Technology, Beijing University of Agriculture/Beijing Key Laboratory of New Agricultural Technology in Agriculture Application/National Demonstration Center for Experimental Plant Production Education, Beijing 102206, China; 2Key Laboratory for Northern Urban Agriculture of Ministry of Agriculture and Rural Affairs, Beijing University of Agriculture, Beijing 102206, China; 3College of Grassland Science and Technology, China Agricultural University, Beijing 100193, China

**Keywords:** *CNGC*, adverse stress, omics analysis, *G. max*

## Abstract

**Simple Summary:**

*Glycine max* diseases have always been an issue in China, with root rot caused by *Fusarium solani* being the most serious issue in this regard. Here, the mechanism of action of genes showing *F. solani* resistance was studied by analysing the differential expression of the transcriptome of this plant after pathogen infection to provide a theoretical reference for the breeding of disease-resistant soybeans. Thus, four genes that regulate the concentration of inorganic ions inside and outside the membrane via the transmembrane ion channel and participate in stress regulation via signal transduction were identified.

**Abstract:**

Cyclic nucleotide-gated channels (*CNGCs*), non-selective cation channels localised on the plasmalemma, are involved in growth, development, and regulatory mechanisms in plants during adverse stress. To date, *CNGC* gene families in multiple crops have been identified and analysed. However, there have been no systematic studies on the evolution and development of *CNGC* gene families in legumes. Therefore, in the present study, via transcriptome analysis, we identified 143 *CNGC* genes in legumes, and thereafter, classified and named them according to the grouping method used for *Arabidopsis thaliana*. Functional verification for disease stress showed that four *GmCNGCs* were specifically expressed in the plasmalemma during the stress process. Further, functional enrichment analysis showed that their mode of participation and coordination included inorganic ion concentration regulation inside and outside the membrane via the transmembrane ion channel and participation in stress regulation via signal transduction. The *CNGC* family genes in *G. max* involved in disease stress were also identified and physiological stress response and omics analyses were also performed. Our preliminary results revealed the basic laws governing the involvement of *CNGCs* in disease resistance in *G. max*, providing important gene resources and a theoretical reference for the breeding of resistant soybean.

## 1. Introduction

*Glycine max*, an important grain, oil source, and feed crop worldwide, has always attracted considerable interest globally. Its market demand directly reflects the living standards of residents, and it plays an important role in meeting grain, oil, and protein needs [1,2]. However, during its growth and development, it faces various adverse stresses, which considerably affect its yield and quality. This situation can be primarily attributed to the fact that the growth and development stages of *G. max* in China are dominated by a mild and rainy climate, with frequent diseases, root rot being the most serious issue in this regard [3].

Ca^2+^ transduction through Ca^2+^ channels is an important regulatory pathway in plants during response to stress. To maintain physiological activity in the face of adversity, plant surface cells pump Ca^2+^ ions from locations with high concentrations to locations with low concentrations through Ca^2+^ channels. Thus, the Ca^2+^ ion concentration in the membrane increases instantaneously and coupled with the duration and amplitude of this process, calcium signals are generated. Subsequently, signal transmission and body regulation are realised via the conversion of these chemical and electrical signals. Alternatively, the Ca^2+^ ions can combine with calmodulin (CaM) to form the Ca^2+^-CaM complex, which directly acts on bodily regulatory systems.

Cyclic nucleotide-gated channels (CNGCs), which are nonselective cation channels located on the plasma membrane, are important pathways for conducting Ca^2+^ ions in signal transduction [4]. Plant *CNGCs* are composed of six transmembrane domains (S1–S6), a pore region between the fifth and sixth domains (P), and C-terminal CaMB and CNBD [5]. Specifically, CNBD, the most conserved region of CNGCs, consists of the spanning phosphate-binding cassette (PBC) and the “hinge” region (adjacent to the PBC). Notably, PBC binds to the sugar and phosphate groups of cyclic nucleotide ligands [6], and the “hinge” region contributes to ligand binding efficacy and selectivity [7]. Studies have also shown that the carboxyl tails of *CNGCs* in several plants bind CaM in a Ca^2+^ dependent manner [8,9,10,11]. Further, the *CNGC* gene family plays an irreplaceable role in plant growth and development, as well as in plant abiotic and biological stress response processes. For example, the *CNGC* gene family is involved in seed germination during plant growth and development. It is also essential for polarized tip growth of pollen [12], root hair tip growth [13], leaf senescence [14], and rhizobia and mycorrhizal symbiosis in leguminous roots [15]. Additionally, it plays an important role in plant response to salt stress [16], heat stress [17], drought stress [18], heavy metal ion stress [19], and coping with pathogen invasion [20,21].

Even though the *CNGC* gene family in many plants has been identified, its role in *G. max* has not yet been sufficiently studied. Therefore, in the present study, we identified the *CNGC* gene family of *G. max*, and investigated its evolution and development. We also identified the whole genome of the *CNGC* gene family of six other legumes (*Medicago truncatula*, *Cicer arietinum*, *Cajanus cajan* (pigeon pea), *Lotus japonicus*, *Vigna angularis*, and *Phaseolus vulgaris*) that have been sequenced. Subsequently, the *G. max CNGC* genes involved in two kinds of disease stress responses were screened via the assessment of *G. max* disease stress and gene expression status. We believe that the results of this study will serve as a foundation for further research and provide direction for the subsequent breeding of stress resistant *G. max*.

## 2. Materials and Methods

### 2.1. Identification and Phylogenetic Analysis

To investigate the evolutionary relationships among members of the *CNGC* gene family in *G. max*, we used seven leguminous species. The protein sequences of *Arabidopsis thaliana*, *G. max*, *M. truncatula*, *P. vulgaris*, and *C. cajan* (pigeon pea) were obtained from the Ensembl database (http://plants.ensembl.org/index. html accessed on 2 February 2023), while those of *L. japonicus*, *V. angularis*, *C. arietinum* were obtained from the GigaDB dataset (http://gigadb.org/dataset/100028 accessed on 2 February 2023). Further, the *CNGC* protein sequences of *A. thaliana* was used as the query object, and the potential candidate genes were obtained by performing a local BLASTP search, considering an E value of 1 × 10^−10^. The protein sequences were then uploaded to the Conserved Domain Database (CDD) (http://www.ncbi.nlm.nih.gov/Structure/cdd/wrpsb.cgi/ accessed on 1 March 2021), SMART database (http://smart.embl-heide lberg.de/ accessed on 3 March 2021), and PFAM database (http://pfam.xfam.org/ accessed on 4 March 2021). Finally, *CNGC* genes containing two basic domains and one *CNGC*-specific motif were identified in the genomes of the seven legumes [22,23,24,25,26].

To analyse the evolutionary relationships among the members of the *CNGC* gene family in *G. max*, we used the NJ method of MEGA-X, set the bootstrap value at 1000, and constructed a phylogenetic tree based on the full-length sequence of the *CNGC* protein in *A. thaliana* and the seven other legumes [22,23,24].

### 2.2. Location of CNGC Genes on Chromosomes and Conserved Motifs

The locations of each *CNGC* gene on 20 *G. max* chromosomes were determined using the Ensembl plant database, and the distribution information map of the genes on these chromosomes was established using MapChart. Further, the MEME online program (http://meme.nbcr.net/meme/intro.html accessed on 23 May 2021) for protein sequence analysis was used to identify the conserved motifs in the *CNGC* proteins in the leguminous plants [23,27].

### 2.3. Collinear Analysis of CNGC Genes

The non-synonymous substitution rate (Ka), synonymous substitution rate (Ks), and the Ka/Ks ratio were calculated using KaKs Calculator v2.0. Further, the NG method was used to study dispersion [28], and divergence time (T) was calculated using the formula: T = Ks/(2 × 6.1 × 10^−9^) × 10^−6^ million years ago (MYA) [29,30,31]. To further infer the phylogenetic mechanism of *G. max* and the other selected species and obtain a collinear relationship between homologous *CNGC* genes, MCScanX software was used to construct collinear analysis diagrams [32].

### 2.4. Analysis of Cis-Acting Element

To clarify the possible regulatory mechanism of *CNGC* genes in abiotic or biological stress responses, we used PlantCARE (https://bioinformatics.psb.ugent.be/webtools/plantcare/html/ accessed on 24 May 2021) to analyse the promoter sequences online and determined the cis-regulatory elements in the promoter region within 1500 bp upstream of the start codon of the coding region of *CNGC* genes [33].

### 2.5. Pathogen Isolation, Purification, and Morphological Identification

Seedlings with root rot were selected for pathogen isolation and culture. Specifically, diseased *G. max* root samples were isolated using the tissue isolation method. Thereafter, the root area around the boundary between the diseased and healthy area was cut using scissors and treated with 75% ethanol for 30 s followed by NaClO treatment for 3 min [34]. This was followed by rinsing thrice with sterilized water and drying on sterilised filter paper. Next, the samples were placed on a potato dextrose agar plate and cultured upside down in a constant temperature incubator at 28 °C after sealing [34,35]. After the colonies had grown, isolated pathogens were purified via single-spore isolation on the PDA plates [34]. Next, the morphological characteristics of the pathogens were observed regularly until the colonies grew to fully occupy the PDA plate. The morphological identification of the pathogens was conducted considering the morphology of the colonies and spores.

### 2.6. Pathogenicity Detection

The pathogenicity of the isolated fungi was determined according to Koch’s law. The cultivar tested in this regard was “*Zhonghuang 13*”. In brief, *G. max* seedlings of similar size and health status were selected and disinfected using 75% alcohol for subsequent use. Next, the fungus was inoculated with PDA culture medium, and after the fungal colonies grew to occupy 2/3 of the dish, they were crushed via a hole punch. Next, they were placed in a conical flask containing 100 mL of PDB medium and incubated at 25 °C for 5–7 days at a rotation speed of 180 rpm. Microscopic examination was then performed every other day to observe sporulation. At the end of this incubation period, a conidial suspension was prepared and centrifuged. The sediment thus obtained was then diluted with sterile water to a spore concentration of approximately 5 × 10^5^ CFU/mL. *G. max* seedlings were then immersed in spore suspension and observed daily. Conversely, the seedlings infused with the PDB medium were cultured in a constant temperature incubator at 28 °C. The disease condition of the root system was observed daily, and photos were taken for subsequent analysis. After disease onset, the diseased tissue was isolated and cultured again to determine whether the same pathogen was present [35].

### 2.7. Molecular Identification of Pathogenic Fungi

Pure cultured hyphal DNA obtained via isolation and purification was extracted using a fungal genomic DNA rapid extraction kit and used as a template for the PCR amplification of ITS gene fragments with primers, ITS1/ITS4. The reaction conditions were as follows: pre-denaturing at 95 °C for 4 min, denaturing at 95 °C for 30 s, annealing at 56 °C for 30 s, extending at 72 °C for 45 s, for 35 cycles, and finally, extending at 72 °C for 7 min [36,37,38]. The PCR amplification products were detected via 1% agarose gel electrophoresis and sequenced by Beijing Bomaide Gene Technology Co., Ltd. (Beijing, China). The taxonomic status of the pathogens was determined via BLAST analysis of the NCBI database (https://www.ncbi.nlm.nih.gov/ accessed on 5 August 2021), followed by the downloading of relevant crop sequences based on alignment, and finally, the construction of a phylogenetic tree using the adjacency method of MEGA-X (https://www.megasoftware.net/ accessed on 5 August 2021). Thus, the taxonomic status of the pathogens was determined.

### 2.8. Pathogen Inoculation and Sampling Time

The *Zhonghuang 13* cultivar was selected as the test plant. Specifically, pathogens were inoculated via root irrigation after the trilfoil period of the plant. After inoculation for 1, 3, and 7 days, root samples were selected for transcriptome detection and analysis [39].

### 2.9. Screening of Differentially Expressed Genes (DEGs)

To study the changes in gene expression in the different samples or groups, DEGs owing to pathogen stress were identified. Further, the expression matrix of all the samples at the gene level was obtained based on the quantitative analysis of gene expression data. The DEGs were analysed using the edgeR package. The input data for differential gene expression analysis was the read count data obtained from gene expression level analysis. This data analysis process was divided into three main steps, namely, read count normalization, the calculation of hypothesis testing probability (*p*-value) according to the model, and multiple hypothesis testing based on the obtained *p*-value to obtain the q-value (multiple hypothesis testing value) [40].

### 2.10. Analysis of Enrichment

TopGO software was used to conduct GO enrichment analysis for the DEGs, count the number of genes associated with each GO term that showed significant enrichment, and conduct secondary classification statistics. The first 20 terms significantly enriched were selected and displayed using a bar graph [41]. Further, KOBAS (V3.0) software was used for KEGG enrichment analysis. Thus, the first 20 significantly enriched pathways (less than 20% of all the enriched pathway terms) were selected for statistical analysis [42].

### 2.11. Protein–protein Interaction (PPI) Prediction

To clarify whether the screened DEGs interacted with stress-related genes, PPI analysis was performed using the interaction relationships in the STRING protein interaction database (http://string-db.org/ accessed on 5 January 2022) [43].

## 3. Results

### 3.1. Identification and Phylogenetic Analysis of the CNGC Gene Family

Based on 20 *A. thaliana CNGC* protein sequences, we identified 143 *CNGC* genes containing two basic domains (PF00520/PF07885 and PF00027) and one *CNGC*-specific motif in legumes (Table 1). The 143 CNGC genes identified were distributed as follows: 35 from *G. max*, 17 from *C. arietinum*, 11 from *L. japonicus*, 19 from *M. truncatula*, 22 from *P. vulgaris*, 21 from pigeon pea, and 18 from *V. angularis*. The use of ExPASy (http://prosite.expasy.org/ accessed on 4 March 2021) showed that the 143 *CNGC* proteins had different physical and chemical properties (Appendix A). Their amino acid-lengths varied from 418 (*PpCNGC19*) to 778 (*MtCNGC18*), their molecular weights varied from 5.6 kDa (*PpCNGC5*) to 91.1 kDa (*GmTCP39*), and their isoelectric points varied between 7.9 (*CaCNGC4*) and 10.37 (*LjCNGC9*, *ATCNGC19*).

To analyse the evolutionary relationships among the genes in the *CNGC* family of *G. max*, we constructed a phylogenetic tree based on the structure and classification system used for *A. thaliana* sequences (Figure 1). Thus, considering the seven legumes involved in this study, we observed that the genes predominantly belonged to Group III. Specifically, *G. max*, *C. arietinum*, *L. japonicus*, *M. truncatula*, *P. vulgaris,* pigeon pea, and *V. angularis* accounted for 28.57%, 29.41%, 27.27%, 36.84%, 31.82%, 33.33%, and 38.89% of the evolutionary relationships, respectively. Further, relative to the genes in the *CNGC* family for *G. max* and *L. japonicus*, those in the other five legumes were similar, but different from those two. In general, the *CNGC* protein family of *P. vulgaris* showed a close relation with that of *V. angularis*. Further, those of *C. arietinum* and *G. max* showed close relationships with those of *M. truncatula* and pigeon pea, respectively.

### 3.2. Distribution, Gene Structure, and Motif Analysis of GmCNGC Genes on Chromosomes

Chromosomal localisation analysis (Figure 2) showed that the *CNGCs* of *G. max* are located on 14 chromosomes (i.e., on chromosomes 2, 3, 4, 6, 7, 8, 9, 12, 13, 14, 16, 17, 18, and 19), and their distribution was uneven. Some chromosomes (12 and 16) carried five genes, while the others (2, 3, 7, 8, 13, 14, 17, 18, and 19) carried fewer genes. Further, chromosomes 1, 5, 10, 11, 15, and 20 did not harbour any *GmCNGC* genes. Our results also indicated that most of the *GmCNGC* genes were distributed at the end or proximal part of the chromosomes. However, we observed that *GmCNGC18*, *GmCNGC27*, *GmCNGC10*, and *GmCNGC16* on chromosomes 6, 8, 9, and 13, respectively, are located in the middle of the chromosomes.

A phylogenetic tree was constructed with the conserved motifs using MEME (Figure 3). Comparative analysis of the phylogenetic tree and conserved motifs (Figure 3) showed that the conserved motifs of members of the same group were similar, and the conserved motifs of homologous genes were highly consistent. Additionally, the number of conserved motifs in each group was approximately the same (average of seven motifs). However, some of these were unique to Group IV-A, e.g., compounds 20, 4, and 2. Additionally, the number of introns varied in the range 5–12; however, the distribution of exons and introns in the clustered genes was very similar. We also observed that the intron regions corresponding to group Ⅳ-B were longer than those corresponding to the other groups were.

### 3.3. Collinearity and Evolutionary Analysis of CNGC Genes

The divergence times of the Gm-Lj, Gm-Mt, Gm-Pp, Gm-Pv, Gm-Va, and Gm-Ca homologous pairs were 29–88, 9–88, 12–81, 16–101, 18–97, and 26–85 MYA, respectively. The divergence times for the Lj-Mt, Lj-Pp, Lj-Pv, and Lj-Va homologous pairs were 34–93, 33–97, 30–97, and 32–88 MYA, respectively. Further, the Pp-Mt, Pp-Pv, and Pp-Va pairs were separated by 30–96, 21–96, and 20–78 MYA, respectively, and the divergence times of Pv-Mt, Pv-Va, and Va-Mt are 33–90, 6–91, and 36–94 MYA, respectively. Thus, the divergence times of the seven legumes were similar. Notably, the earliest and latest divergence times (16–101 and 20–78 MYA, respectively) corresponded to the Gm-Va and Pp-Va homologous pairs, respectively. Moreover, most of the Ka/KS values were below 0.4, indicating that all the legumes considered in this study show strong purifying selection (Appendix A)

To further infer the phylogenetic mechanisms of *G. max* and the other selected species, and to obtain collinear relationships between homologous *CNGC* genes, collinear analysis diagrams were constructed (Figure 4). The results thus obtained showed the existence of a one-to-one correspondence among all the *CNGC* genes in the seven legumes. Further, among the 35 *GmCNGC* in *G. max*, the number of *CNGC* gene homologous pairs to *M. truncatula*, *C. arietinum*, pigeon pea, *L. japonicus*, *V. angularis*, and *P. vulgaris* were 9, 18, 15, 3, 15, and 22, respectively. This phenomenon indicated less *G. max* and *V. angularis* genomic rearrangements after the lineages diverged and more *G. max* and *P. vulgaris* genomic rearrangements before lineage divergence. Additionally, *M. truncatula*, *P. vulgaris*, and pigeon peas showed few genomic rearrangements after divergence.

### 3.4. The Cis-Acting Element in the CNGC Gene Promoter Sequence

Our results showed that *CNGC* contained at least 30 cis-regulatory elements, including drought, high temperature, low temperature, osmotic stress, light, phytohormone (auxin, abscisic acid, gibberellin, MeJA, and ethylene), pathogen, and damage response elements. From Figure 5, which shows the main cis-regulatory elements in the *G. max CNGC* gene, it is evident that all the *CNGC* genes in *G. max* contained abscisic acid response-related elements, ABRE and ABRE4, light-response elements, ACE and AAAC-Motif, AAGAA-Motif related to methyl jasmonate response, AC-I and AC-II related to lignin biosynthesis, and ACTCATCCT related to stress response.

### 3.5. Symptoms of Field Diseases

Field observations revealed that root rot, which predominantly affected the root and stem bases of *G. max*, was most severe during the seedling stage of the plant. Further, the root rot primarily started from the fibrils or individual branches of *G. max* plants when the parts above the ground began to wilt. Later, owing to infection by pathogens, the fibrils gradually appeared dark brown and gradually fell off, as infection in the main root progressed. The leaves yellowed upward from the lower end of the morphology; however, the yellowed leaves did not fall off (Figure 6a. Left). In the severe infection cases, the main root completely turned black and showed an ulcerated shape. Further, almost all the fibrils fell off, a few short-branched roots were observed (Figure 6a. Right), and the whole plant appeared necrotic.

### 3.6. Pathogen Isolation, Purification, and Morphological Identification

When collected root tissue samples with similar symptoms were cultured at 28 °C in the PDA medium, the colonies grew faster and could cover the entire discs within approximately 5 days. The colonies appeared white at the early stage, and at the later stage, some produced pigments and appeared red (Figure 6b). Further, the colonies were round or nearly round, and their edges were radial. At the later growth stage, the accurate number of mycelia could be observed under a light microscope. Thus, we noted that the conidia were all standard sickle-shaped with a septum in the middle. Most of the conidia had three septa, while a few had 5 septa. The size of the conidia was approximately 15–40 μm × 4–6 μm (Figure 6c), and the pathogen could easily sporulate. The spore shape and colony morphology were consistent with the morphological characteristics of *F. solani* as described by Song [44]; therefore, the pathogen was preliminarily identified as *F. solani*.

### 3.7. Pathogenicity Detection

Pathogenicity was detected indoors. Specifically, the isolated and purified pathogens were inoculated via root irrigation. Relative to the observations made at 0 h (Figure 7a), after 24 h, the *G. max* root system changed slightly (Figure 7b), and after 72 h, the number of lateral roots decreased significantly, and the main root and stem base became brown (Figure 7c). Further, after 7 days, the fibrous roots decayed and fell off, and the branched roots became increasingly shorter owing to decay. The entire main root and stem base appeared black, and the cortex showed ulceration. The symptoms of the disease were the same as those of root rot under natural conditions (Figure 7d), conforming Koch’s law and indicating that this bacterium is pathogenic to *G. max*. Simultaneously, based on statistical analysis, the incidence of pathogen re-infection was 100%, indicating that this fungus is highly pathogenic to *G. max* leaves.

### 3.8. Molecular Identification and Analysis of Pathogen

The isolated pathogens were extracted and tested for DNA, and their sequences were compared using the NCBI database. The result (Figure 8) showed that we observed 99.81% consistency with the gene sequence of *F. solani* (Sequence ID: MN216225.1) based on comparison with the gene sequences of other Fusarium species. Thus, a phylogenetic tree was constructed. Combined with the results of morphological identification, we observed that the pathogenic fungus was closely related to *F. solani*, confirming it as the pathogen.

### 3.9. Screening of DEGs after Disease Stress

The results (Figure 9) showed that four of the 35 *CNGC* genes were abnormally expressed during stress treatment. The stress treatment group showed a downregulation in the transcriptional level of *GmCNGC22* by day 3, while *GmCNGC27* was significantly upregulated by day 7. To determine the responses of the *CNGCs* before and after infection, we analysed the transcriptome of *GmCNGC5* at 1 d and 7 d after stress exposure and screened the DEGs. Thus, we observed a significant downregulation in the transcriptional level of *GmCNGC5*, while that of *GmCNGC31* was upregulated.

### 3.10. GO and KEGG Enrichment Analysis

GO enrichment analysis (Figure 10) showed 20 major biological processes (BP), among which there were more enriched differential genes in response to stimuli (GO:0010052), chemicals (GO:0008284), and stress (GO:0015985). Among the selected DEGs, *GmCNGC5*, *GmCNGC22*, *GmCNGC27*, and *GmCNGC31* were mainly enriched in response to stress (GO:0015985). Further, regarding molecular function (MF) categories, several DEGs were enriched in oxidoreductase activity (GO:0016491) and cofactor binding (GO:0048037), whereas *GmCNGC5*, *GmCNGC22*, *GmCNGC27*, and *GmCNGC31* were mainly enriched in transmembrane transporter activity. Regarding cellular components (CC), the DEGs were mainly concentrated in the cell periphery (GO:0071944). *GmCNGC5*, *GmCNGC22*, *GmCNGC27*, and *GmCNGC31* were mainly enriched in the cell periphery (GO:0071944) and plasma membrane (GO:0044459) (Appendix A).

KEGG analysis (Figure 11) showed that the signalling pathways associated with the relevant DEGs following disease stress mainly included protein processing in the endoplasmic reticulum, tyrosine metabolism, fatty acid degradation, alpha-linolenic acid metabolism, glycolysis/gluconeogenesis, spliceosome metabolism, plant–pathogen interaction, endocytosis, biosynthesis of secondary metabolites, and metabolic pathways. Further, under stress, the DEGs in the *CNGC* gene family showed enrichment in plant–pathogen interaction pathways.

### 3.11. Protein Interaction

To identify proteins interacting with the screened *CNGC* in *G. max*, we constructed a PPI network diagram as shown in Figure 12. This figure shows interaction between *GmCNGC5* and JASMONATE ZIM (JAZ) domain-containing proteins. Studies have also shown that in plants, JAZ family proteins play a crucial role in response to environmental stress [45]. *GmCNGC22* interacted with GF14C, a protein with important functions in defence and abiotic stress response [46]. Further, *GmCNGC27* interacted with GMERA1A (protein farnesyltransferase) and GMERA1B, which reportedly, can respond to *G. max* stress by regulating ABA signalling in guard cells [47]. Further, *GmCNGC31* interacted with Kup system potassium uptake proteins and protein detoxification 44.

## 4. Discussion

*CNGC* gene families have been reported in several plants, including *A. thaliana* [10], rice [48], tomato [49], pear [50], wheat [51], and Chinese cabbage [22]. However, most of these previous studies were based on the evolutionary analysis of the *CNGC* gene family of a single species. Further, studies on the analysis and comparison of CNGC gene families from species of the same family and genus as *G. max* are rare. Moreover, few studies have compared and screened *G. max* gene response levels under biological and abiotic stresses. In this study, to discuss the evolutionary relationship of the *CNGC* gene family in *G. max*, we identified and analysed the *CNGC* gene families of *G. max*, *C arietinum*, *L. japonicus*, *M.* truncatula, *P. vulgaris*, pigeon pea, *and V. angularis*, which contained 35, 17, 11, 19, 22, 21, and 18 *CNGC* genes, respectively, with two basic domains (PF00520/PF07885 and PF00027) and one *CNGC*-specific motif. Based on our analysis, it was obvious that the *CNGC* genes of the other six legumes, except *G. max* are very similar in number. This may be related to whole-genome replication, which is common in angiosperms [52]. Further, we divided all the *CNGC* gene families into four groups. Groups I–III were monogenic, while group IV could be further divided into two distinct branches (groups IV-A and IV-B). We also observed that for the seven legumes, Groups III and II contained the largest and smallest proportions of genes, respectively. Subsequently, the 35 *CNGC* genes in *G. max* were named *GmCNGC1* to *GmCNGC35* (Appendix A) in sequence according to the grouping method used for *A thaliana*. The sequences of *G. max CNGC* proteins in the evolutionary tree and *CNGC* gene families of other legume crops were named in a similar manner. In general, the *CNGC* protein family of *G max* showed a close relationship with that of pigeon peas, and to measure the evolutionary pressure on the coding sequences, the Ka/Ks ratio was used. The Ka/Ks ratio obtained in the present study indicated that the *CNGC* genes in the seven legumes have strong selection constraints. Further, our analysis of gene structure showed that the conserved motifs of members of the same *CNGC* gene family were similar. The conserved motifs of the homologous genes were also highly consistent. This is in agreement with the results reported by Qingqing [22]. Furthermore, most of the conserved motifs, e.g., motifs 20, 4, and 2, were unique to Group IVA; therefore, some functions of the *CNGCs* genes in Group IV-A were not observed in the other groups. Conserved motifs 19, 7, 18, and 14 were common to almost all the *CNGCs*.

The isoelectric point (pI) and electric charge of a protein, which are important factors that affect its solubility, subcellular localisation, and interactions, depend on the insertion and deletion of homologous sites and the ecology of the organism. Reportedly, when cytoplasmic proteins have an acidic pI (pI < 7.4), their nuclear counterparts tend to have a more neutral pI (7.4 < pI < 8.1), while the pI of the membrane becomes more alkaline. Therefore, alkaline residues located on both sides of the membrane span influence the stability of the proteins in the membrane [53]. The net charge of proteins is a basic physical property, and its value directly affects their solubility, aggregation, and crystallisation. The CNGCs located on the membrane have physicochemical properties that are considerably different, and theoretically, they basically function as buffers. These differences reflect changes in protein composition, which affects the associations between receptor-charged ligands, folding and stability, dissolution and precipitation, and selective ion transport in protein channels. Interestingly, our prediction of *AtCNGC19* and *AtCNGC20* was based on the chloroplast membrane; however, previous studies have confirmed that *AtCNGC19* and *AtCNGC20* in Arabidopsis are not confined to the chloroplasts. In contrast, *CNGC19* FL-GFP and *CNGC20* FL-GFP were simultaneously expressed in protoplasts with vacuole membrane marker (α tip m Cherry and γ tip m Cherry) co-location. During the lengthy process of biological evolution, plants acquire complex gene regulation mechanisms to mitigate the effects of adverse environments. In our study, most *CNGC* showed association with stress-related regulatory elements, such as drought, high temperature, low temperature, pathogenic bacteria, injury, and other related elements. The discovery of these elements highlighted the possibility to conduct subsequent experiments and indicated that it is necessary to perform further experiments for the functional verification of *CNGCs*.

Morphological and molecular identification of root rot occurrence and severity in *G. max* growing areas in China indicated that the pathogen causing this disease is predominantly *F. solani.* It has also been demonstrated that *F. solani* hosts comprise most terrestrial plants of the orders Solanaceae [54,55], Rutaceae [56], Apiaceae [57], Cucurbitaceae [58], Araliaceae [59] and Araceae [60]; therefore, it has a wide range of hosts, and the main infestation site is the root system and stem base of the plants. It can also cause large-scale outbreaks of root rot under suitable conditions of temperature and humidity, and because of its wide host range and strong pathogenicity, it seriously restricts local agricultural production and development.

Recent studies have shown that *CNGC* genes can regulate intracellular Ca^2+^ concentration based on interactions between gene families and multiple transcription factors. These genes also play an important role in plant responses to various environmental stresses, such as saline-alkali, drought and flood, cold and heat, pathogen invasion, and heavy metals exposure [61,62,63,64,65,66]. In this study, we observed *GmCNGC* gene responses to disease stress. Specifically, under disease stress, *GmCNGC27* and *GmCNGC31* were significantly upregulated, whereas *GmCNGC5* and *GmCNGC22* were significantly downregulated. Further, via the functional enrichment analysis of the DEGs, we observed that *CNGC* gene family pathways are involved in stress regulation following plant pathogen interaction. These genes act on the plasma membrane, regulate the concentration of inorganic ions inside and outside the membrane via transmembrane ion channels, and intervene in body regulation under stress through signal transduction, consistent with the previously reported observation that *CNGC* functions as a ligand-gated channel [67].

Overall, these abovementioned findings indicated that the genes of the *CNGC* gene family might participate in both biological stress and abiotic stress. Therefore, to a certain extent, this study provides genetic resources and can serve as a theoretical reference for further studies on the breeding of stress resistant *G. max*.

## 5. Conclusions

In this study, we identified 35 *GmCNGC* genes in the *G. max* genome. Phylogenetic analysis revealed that all these *GmCNGCs* contain two basic domains and one *CNGC*-specific motif. Further, a close relationship was observed between the *CNGC* gene family of *G. max* and that of *C. cajana*. Disease stress tests showed that *GmCNGC5*, *GmCNGC22*, *GmCNGC27*, and *GmCNGC31* could respond to disease stress by regulating the concentration of inorganic ions inside and outside the cell membrane via transmembrane ion channels to achieve signal transduction and participate in the stress regulation system.

## Figures and Tables

**Figure 1 biology-12-00439-f001:**
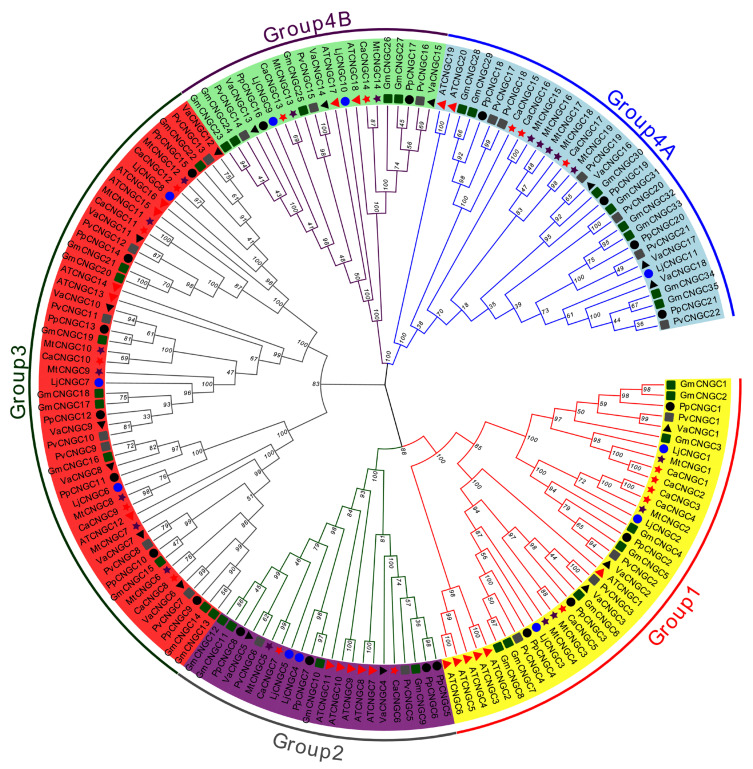
Systematic evolutionary relationships within the CNGC gene family (AtCNGC, GmCNGC, MtCNGC, CaCNGC, PpCNGC, LjCNGC, VaCNGC, and PvCNGC refer to the gene families of *A. thaliana*, *G. max*, *M. truncatula*, *C. arietinum*, *C. cajan* (pigeon pea), *L. japonicus*, *V. angularis*, and *P. vulgaris*, respectively).

**Figure 2 biology-12-00439-f002:**
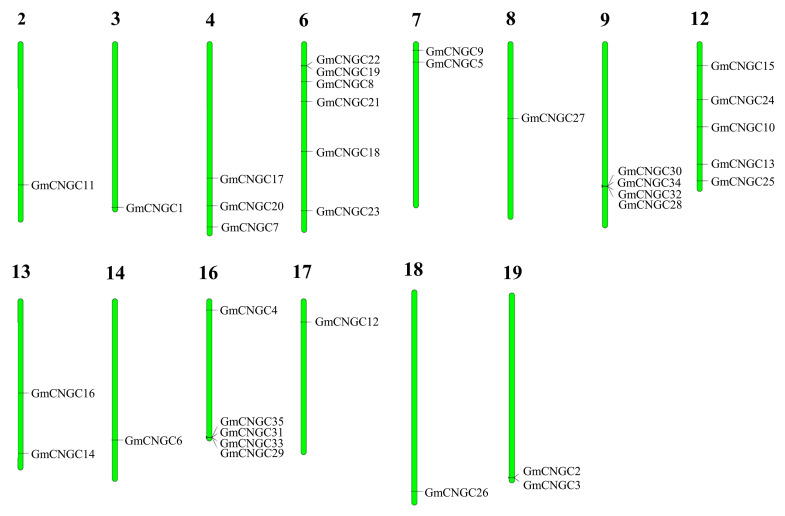
Locations of *CNGC* genes on *G. max* chromosomes.

**Figure 3 biology-12-00439-f003:**
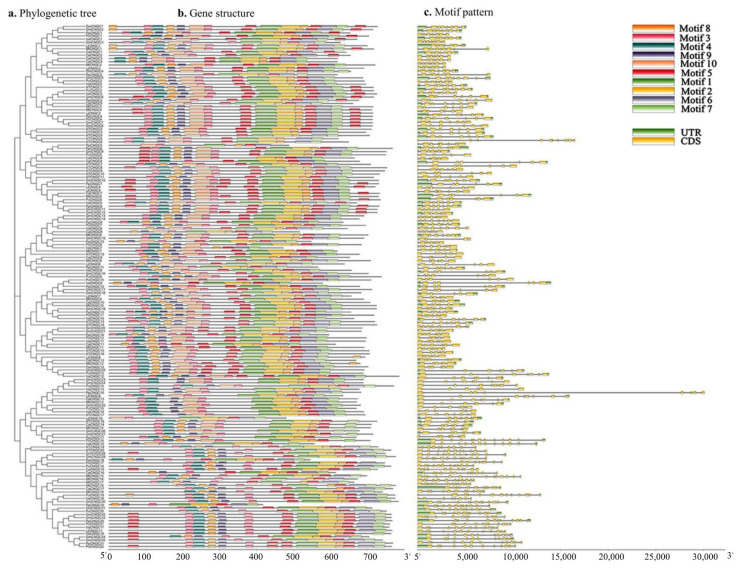
Phylogenetic relationships, gene structure, and architecture of conserved protein motifs in *CNGC* genes from legumes.

**Figure 4 biology-12-00439-f004:**
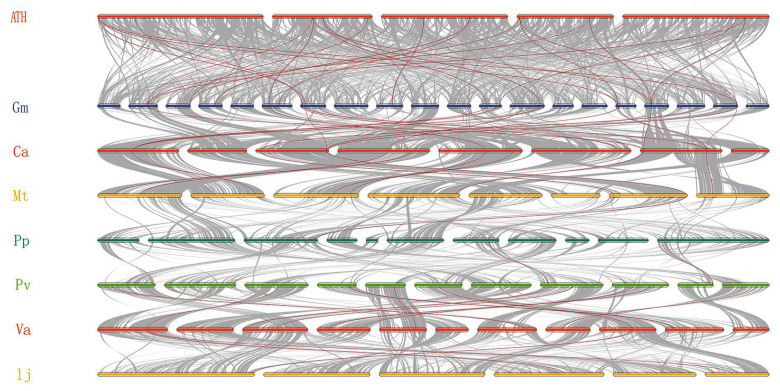
Collinear analysis of *CNGC* genes in *G. max*, *A. thaliana*, and six other legumes (The grey lines in the background indicate collinear regions in the genomes of *G. max* and other plants, while the red lines highlight collinear *CNGC* gene pairs. The prefixes ATH, Gm, Mt, Ca, Pp, Lj, Va, and Pv refer to *A. thaliana*, *G. max*, *M. truncatula*, *C. arietinum*, pigeon pea, *L. japonicus*, *V. angularis*, and *P. vulgaris*, respectively).

**Figure 5 biology-12-00439-f005:**
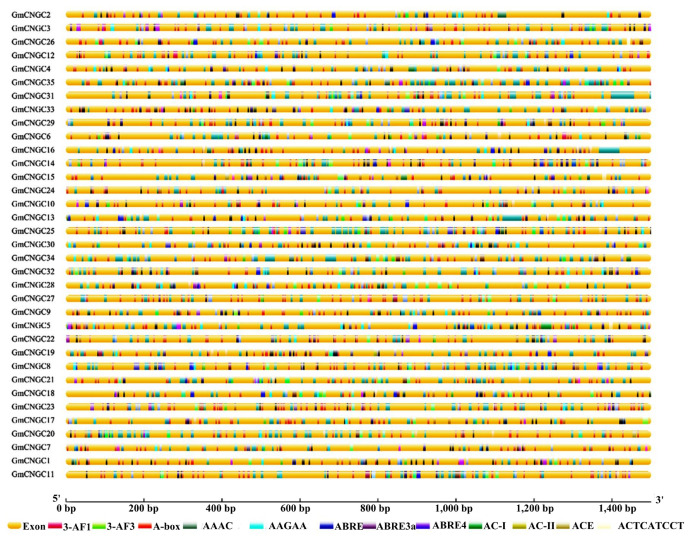
Cis-acting elements of the promoter sequence of *CNGC* genes in *G. max*.

**Figure 6 biology-12-00439-f006:**
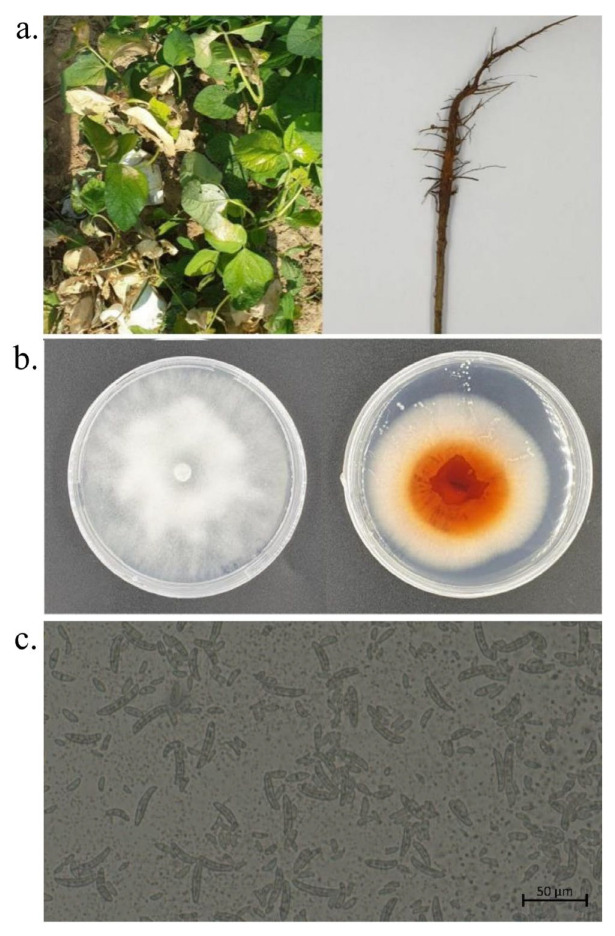
Symptoms of disease in the field and the morphological characteristics of the pathogens ((**a**) symptoms in the field; (**b**) colonies of the pathogen isolated from the PDA medium; (**c**) conidia shape under the microscope).

**Figure 7 biology-12-00439-f007:**
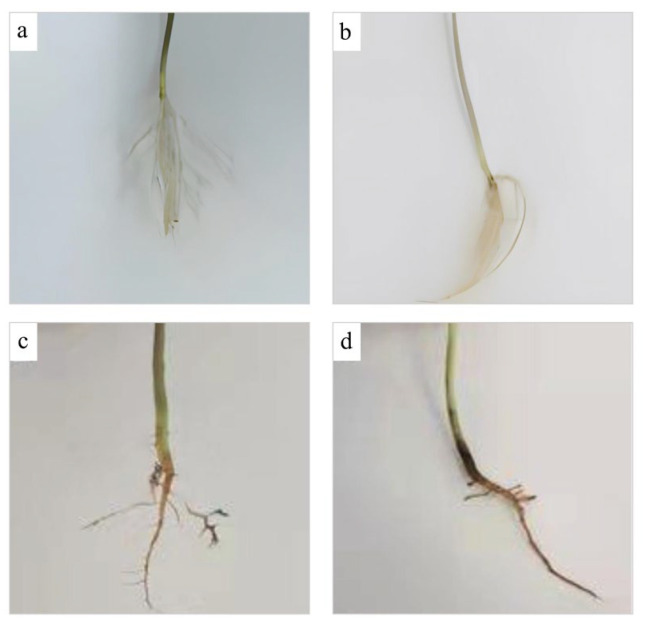
Root changes in *G. max* seedlings after inoculation with pathogenic fungi at (**a**) 0 h, (**b**) 24 h, (**c**) 3 d, and (**d**) 7 d.

**Figure 8 biology-12-00439-f008:**
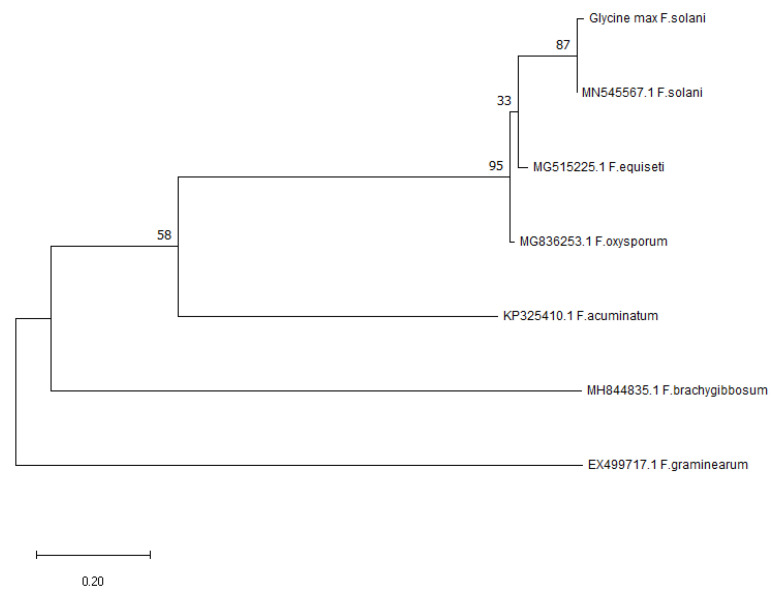
Phylogenetic tree based on the ITS-rRNA of the pathogenic fungus and other phytophysical fungi. Note: The data comes from National Centre for Biotechnology Information Search database (NCBI).

**Figure 9 biology-12-00439-f009:**
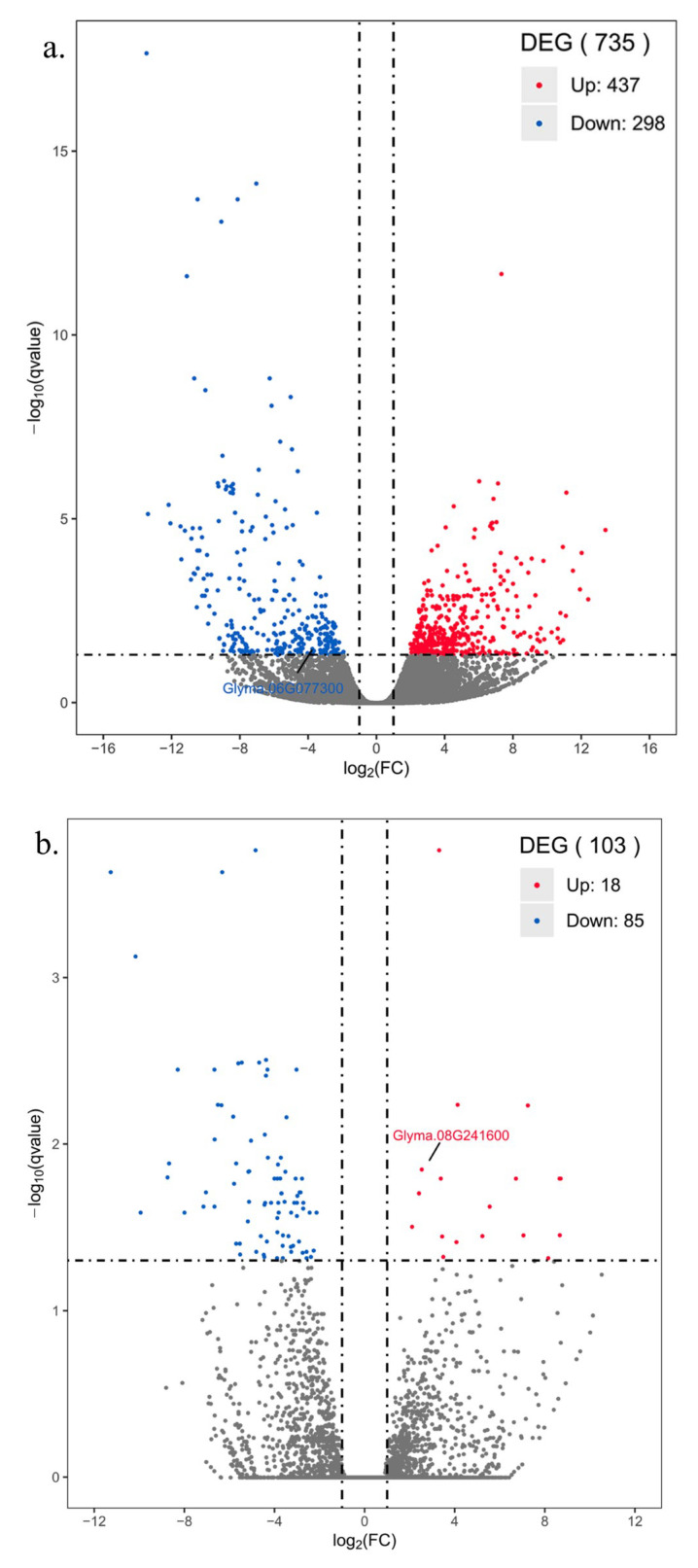
Differences in the transcription levels of *G. max* genes under disease stress (Upregulated and downregulated genes are indicated in red and blue, respectively. Genes with abnormal transcription levels are marked in the figure. (**a**,**b**) represent differences of gene transcription levels between days 3, 7 and blank control, respectively, after disease stress, and (**c**) represent differences of gene transcription levels between days 7 and 1 after disease stress).

**Figure 10 biology-12-00439-f010:**
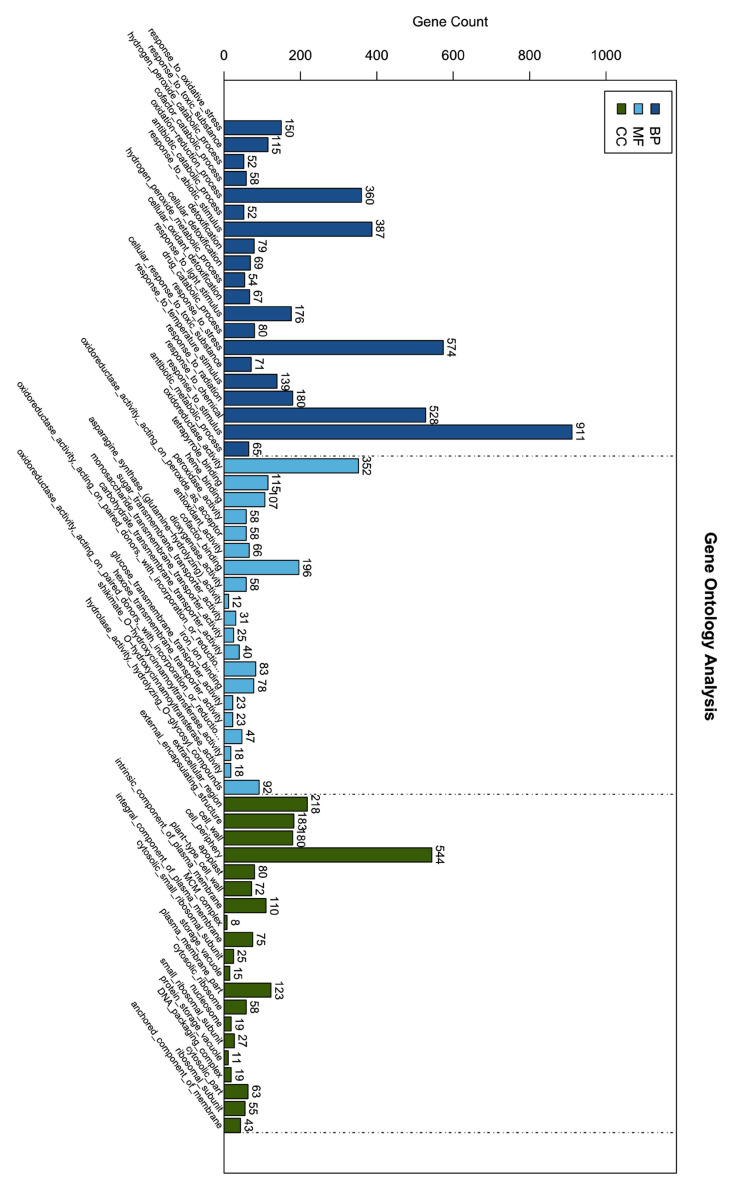
GO enrichment.

**Figure 11 biology-12-00439-f011:**
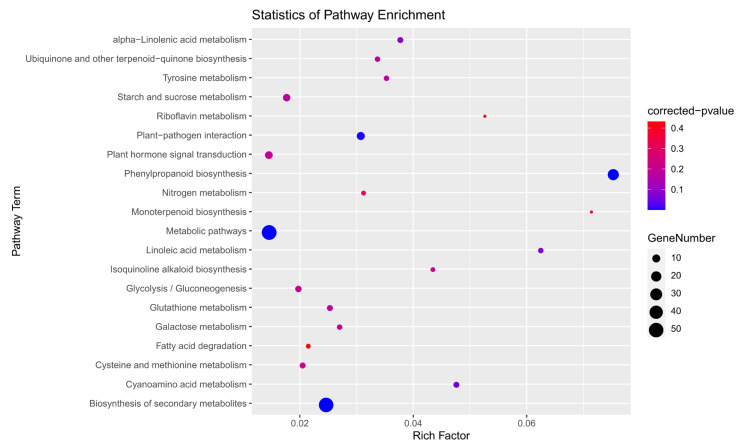
KEGG enrichment.

**Figure 12 biology-12-00439-f012:**
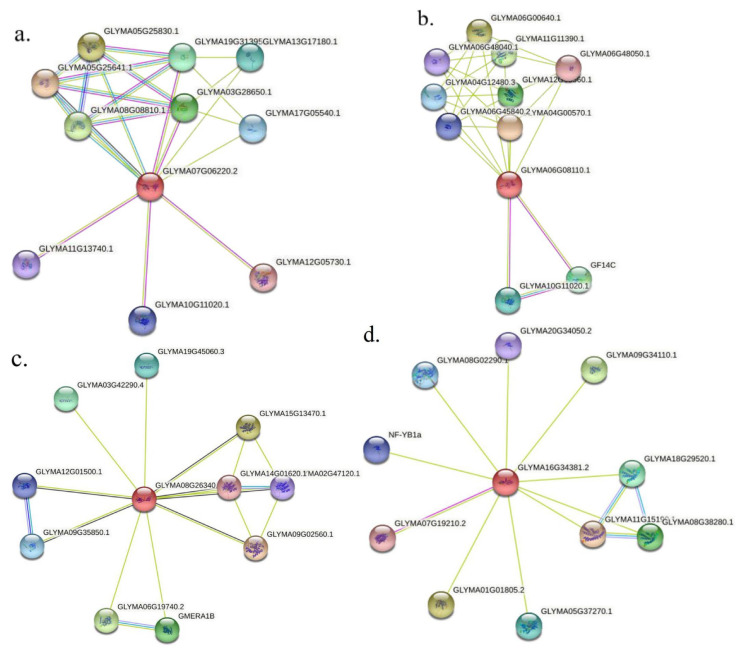
Protein interaction network diagram ((**a**–**d**) represent *GmCNGC5*, *GmCNGC22 GmCNGC27* and *GmCNGC31*, respectively).

**Table 1 biology-12-00439-t001:** Identification and classification of seven leguminous crops.

Species	Number of *CNGC*	Group				
		I	II	III	IV-A	IV-B
*Glycine max*	35	8	4	10	8	5
*Cicer arietinum*	17	5	2	5	3	2
*Lotus japonicus*	11	3	2	3	1	2
*Medicago truncatula*	19	4	1	7	5	2
*Phaseolus vulgaris*	22	4	2	7	6	3
*Cajanus cajan* (Pigeon Pea)	21	4	4	7	4	2
*Vigna angularis*	18	3	2	7	3	3

## Data Availability

Transcriptome data have been submitted to SRA database with the following project number: PRJNA849746. Additionally, the accession number of the pathogen culture is OQ519857. Additionally, other data that support the findings of this study are available from the corresponding author upon reasonable request.

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
