# Peer review of "Identification of *CNGCs* in *Glycine max* and Screening of Related Resistance Genes after *Fusarium solani* Infection"

_biology, 2023, doi:10.3390/biology12030439_

Round 1

Reviewer 1 Report

Major revisions:

1.       The manuscript needs substantial language editing and polishing. There are a lot of misspellings, grammar errors, and inconsistent descriptions, such as terms, names of species.

2.       Section 3.3: The authors need to specify and detail the analysis of homologous and orthologous members of the gene family.

3.       There are too many figures and some of the figures are not informative, I suggested the authors to combine some figures in to one when they are to express different aspects of one thing.

4.       Sometimes the authors used English name of a species, sometimes Latin. They need to link the names when it appear in the manuscript the first time. It’ll make the reviewers and possible the readers to understand.

Minor revisions

Lines 12-18: This section is poorly presented. You need to revise it substantially.

Lines 29-32: The sentence is too long and misleading.

Line 37: Glycine max should be G. max. There are many different spellings, such as Glycine Max, G. Max, the authors need to revise all of them. There are also misuses of the terms of other species.

Lines 46-50: The sentence is too long and unclear.

Line 135: According to your description in this section, I suppose the subtitle of this section is “Screening of differentially expressed genes”.

Line 157: “21 the from Pigeon Pea” “21 from P. pea

Line 158: v. angularisV. angularis

Lines 162-163: Do not need to repeat exactly the gene numbers and the species, as they are listed in the lines 156-157.

Line 184-185: The first sentence “In order to better…” is not necessary.

Line 229: “on the ground” seems to be “above the ground”.

Line 242: “to sporulation” “to sporulate”

Line 243: insert the references.

Author Response

Thank you very much for your letter and advice. I am the first author of “Manuscript ID: biology-2229331”. We have revised the paper, and would like to resubmit it for your consideration. In order to distinguish the revisions after getting the reviewers' suggestions and the revisions in the manuscript due to misspellings, grammar errors, and inconsistent descriptions, I did not mark the suggested revisions’ part, but made point-to-point revisions and detailed revisions records. As for the sentence revisions in the manuscript due to misspellings, grammar errors, and inconsistent descriptions, I have highlighted in red in the revised manuscript. And the detailed correction records of error that you pointed out are as follows: 

Major revisions:

Point 1: The manuscript needs substantial language editing and polishing. There are a lot of misspellings, grammar errors, and inconsistent descriptions, such as terms, names of species.

Response: According to teacher’s suggestion, we proofread the whole article and avoided the above errors.

Point 2: Section 3.3: The authors need to specify and detail the analysis of homologous and orthologous members of the gene family.

Response: According to your suggestion, both figure and article were corrected, proofread and analyzed the section 3.3 again.

Point 3: There are too many figures and some of the figures are not informative, I suggested the authors to combine some figures in to one when they are to express different aspects of one thing.

Response: I corrected and replaced Figure 2, Figure 3, Figure 4, Figure 7, Figure 9, Figure 11 and Figure 13 based on the original data, after gaining teacher’s suggestion. 

Point 4: Sometimes the authors used English name of a species, sometimes Latin. They need to link the names when it appear in the manuscript the first time. It’ll make the reviewers and possible the readers to understand.

Response: In order to facilitate reviewers and possible the readers to understand, I replaced most of English names with Latin names. The English name of Cajanus cajan (Pigeon Pea) is still used because of the problem of abbreviation and CNGC naming, but it was linked the name in each chapter.

Minor revisions

Point 1: Lines 12-18: This section is poorly presented. You need to revise it substantially.

Response: This Simple Summary was revised after teacher’s suggestion.

Point 2: Lines 29-32: The sentence is too long and misleading.

Response: For this error, I rewrote this sentence.

Point 3: Line 37: Glycine max should be G.max. There are many different spellings, such as Glycine Max, G. Max, the authors need to revise all of them. There are also misuses of the terms of other species.

Response: For this question, I checked this article and changed “Glycine max” to “G.max”, but I also remain few names as explanation.

Point 4: Lines 46-50: The sentence is too long and unclear.

Response: Thanks for teacher’s suggestion, this sentence was rewritten.

Point 5: Line 135: According to your description in this section, I suppose the subtitle of this section is “Screening of differentially expressed genes”.

Response: Thanks for teacher’s suggestion, this subtitle of this section was changed.

Point 6: Line 157: “21 the from Pigeon Pea”→“21 from P.pea”

Response: According to teacher’s suggestion, I changed “21 the from Pigeon Pea” to “21 from Pigeon pea”

Point 7: Line 158: v. angularis→V.angularis

Response: Thanks teacher again and thank you very much for your patient review. This error was corrected. 

Point 8: Lines 162-163: Do not need to repeat exactly the gene numbers and the species, as they are listed in the lines 156-157.

Response: The repetitive sentence was deleted. 

Point 9: Line 184-185: The first sentence “In order to better…” is not necessary.

Response: The sentence was deleted. 

Point 10: Line 229: “on the ground” seems to be “above the ground”.

Response: I changed “on the ground” to “above the ground”, after teacher’s suggestion.

Point 11: Line 242: “to sporulation”→“to sporulate”

Response: I changed “to sporulation” to “to sporulate”.

Point 12: Line 243: insert the references.

Response: According to teacher’s suggestion, I reinserted the references.

Reviewer 2 Report

The manuscript is based on identification of the CNGC family genes in soybean involved in disease stress. THe manuscript provides a preliminary idea about the role of CNGC in legumes mainly soybean. However, the manuscript lacks any experimental attributes for CNGC role or selected genes related to resistance. The manuscript needs guidance in language and the manuscript can be improved.

Reviewer comments

1. Line 12: Kindly change this sentence for a better one “Soybean diseases have always been problems in China,”

2. Use italics for nomenclature throughout the manuscript.

3. Materials and methods: Arabidopsis thaliana was also used for the sequence alignment and analysis; however, the writing includes it in the leguminous group, kindly write accordingly. The supplementary file should include the protein IDs for all the legumes used in the analysis. I found only for three legumes in supp table 3. Indicate for the nomenclature of the protein IDs opted in this study, whether it is based on sequence similarity with Arabidopsis homologues.

4. The authors have not shown any experimental evidence supporting the RNA analysis supporting their work.

4. Line 376-378: The sentence is speculative and not based on any experimental evidence however only using in-silico analysis. This work needs further validation. Therefore the sentence needs to be presented as suggestive to the obtained results. IT cannot be shown as conclusive without substantial evidence.

Author Response

Thank you very much for your letter and advice. In order to distinguish the revisions after getting your suggestions and the revisions in the manuscript due to misspellings, grammar errors, and inconsistent descriptions, I did not mark the suggested revisions’ part, but made point-to-point revisions and detailed revisions records. As for the sentence revisions in the manuscript due to misspellings, grammar errors, and inconsistent descriptions, I have highlighted in red in the revised manuscript. And the detailed correction records of error that you pointed out are as follows: 

Point 1: Line 12: Kindly change this sentence for a better one “Soybean diseases have always been problems in China,”

Response: Thanks for teacher’s suggestion, I replaced it with the sentence you provided.

Point 2: Use italics for nomenclature throughout the manuscript.

Response: According to teacher’s suggestion, I corrected the name in the whole article.

Point 3: Materials and methods: Arabidopsis thaliana was also used for the sequence alignment and analysis; however, the writing includes it in the leguminous group, kindly write accordingly. The supplementary file should include the protein IDs for all the legumes used in the analysis. I found only for three legumes in supp table 3. Indicate for the nomenclature of the protein IDs opted in this study, whether it is based on sequence similarity with Arabidopsis homologues.

Response: As for this question, please listen to my explanation: this study was not to classify Arabidopsis thaliana as one of legumes. Due to the family of CNGC genes in legumes is rarely classified and named, therefore, using Arabidopsis thaliana that a model crop commonly used in the study as a reference, seven legumes (Glycine max, Medicago truncatula, Phaseolus vulgaris, Cajanus cajan (Pigeon Pea), Lotus japonicus, Vigna angularis, Cicer Arietinum) and Arabidopsis thaliana CNGC genes are compared. And then the seven legumes are identified, named and classified according to the comparison results. Additionally, the ID number used has been shown in supplementary table 4.

Point 4: The authors have not shown any experimental evidence supporting the RNA analysis supporting their work.

Response: As for this point, I feel very sorry. Due to the influence of COVID-19 pandemic in 2020, the experiment is very difficult, so that I only did partly experiment for evidence supporting the RNA analysis. And now, although COVID-19 pandemic finished, materials lost more and I have also graduated and left the campus. I don't want to waste our hard work, thus, I wrote this article in a hurry. I pray for that I could gain teacher’s forgiveness.

And the results of partly qPCR experiment for evidence supporting the RNA analysis are as follows:

Point 5: Line 376-378: The sentence is speculative and not based on any experimental evidence however only using in-silico analysis. This work needs further validation. Therefore the sentence needs to be presented as suggestive to the obtained results. IT cannot be shown as conclusive without substantial evidence.

Response: According to teacher’s suggestion, I reviewed this paragraph and added "might", "to a certain extent" to increase the stringency of the article.

Reviewer 3 Report

Dear editor and authors,

Thanks for the opportunity to review this interesting manuscript.

Overall, the manuscript provides a comprehensive analysis of CNGC genes in legumes, and the results presented may be useful for future research in this area.

However, it is important to note that the manuscript lacks some key information such as the methods used to perform the analysis.

The text appears to be well-written and comprehensible. There are a few minor grammatical errors and typos, but these do not significantly detract from the clarity of the message. Here are a few examples of corrections that could be made:

Simple Summary:

"related resistance genes were screened and their action mechanism was discussed through the differential expression of transcriptome after pathogen infection" - It would be better to say "the action mechanism of these resistance genes was studied by analyzing the differential expression of the transcriptome after pathogen infection."

"to provide theoretical references for resistance breeding of soybean" - It would be better to say "to provide theoretical references for the breeding of soybean with resistance to diseases."

"The result of this study is that four genes were identified and it was found that the mode of its participation and coordination is to regulate the concentration of inorganic ions inside and outside the membrane through the transmembrane ion channel, and intervene the body regulation in stress by signal transduction." - It would be better to say "The study identified four genes that regulate the concentration of inorganic ions inside and outside the membrane through the transmembrane ion channel and participate in the body's stress regulation through signal transduction."

Introduction

In the first sentence, "in all countries all over the world" is redundant and could be simplified to "worldwide."

In the third sentence, "poor quality of commercial soybeans" should be "poor quality commercial soybeans" (remove the preposition "of").

In the eighth sentence, "body" should be "bodily" for clarity.

MM
In the first sentence, "seven leguminous families" should be changed to "seven leguminous species" to accurately reflect the context.

What is NG method for dispersion?

Line 122: It’s not bacteria, its fungi. "Molecular identification of pathogenic fungi."

In the sentence "To clarify the possible regulatory mechanism of CNGC gene in the abiotic or biological stress response, we use PlantCARE," "use" should be changed to "used" for consistency in tense.

In the paragraph on the isolation, purification, and morphological identification of the pathogen, "PDA culture dish" should be "PDA agar plate" to accurately reflect the context.

In the sentence "Activate the fungi to be cultured = with PDA medium," "Activate" should be replaced with "Inoculate" to accurately reflect the context.

In the sentence "After colonies grew to 2/3 of the dish, bacterial cakes were crushed," "bacterial cakes" should be replaced with "fungal colonies" to accurately reflect the context.

3. Results

You need to describe the methods that were used to reconstruct the phylogeny.

Discussion

In the sentence "35, 17, 11, 19, 22, 21 and 18 CNGC genes containing two basic domains (PF00520/PF07885 and PF00027) and one CNGC-specific motif were identified, respectively," the word "respectively" is used unclear.

In the sentence "Among the seven legumes, the proportion of members in group III was the largest and that in group II was the smallest," the word "that" is missing an antecedent.

In the sentence "The Ka/Ks ratio can be used to measure the evolutionary selection of coding sequences," "evolutionary selection" is not a commonly used term and would be better phrased as "evolutionary pressure."

In the sentence "Interestingly, our prediction of AtCNGC19 and AtCNGC20 was located on the chloroplast membrane, but previous studies have verified that AtCNGC19 and AtCNGC20 in Arabidopsis are not confined to chloroplasts," the phrase "was located" should be "were located" to agree with the plural subject "AtCNGC19 and AtCNGC20."

In the sentence "The discovery of these elements provided a possibility for our subsequent experiments," "our subsequent experiments" is unclear and could be more specific.

Author Response

Thank you very much for your letter and advice. In order to distinguish the revisions after getting your suggestions and the revisions in the manuscript due to misspellings, grammar errors, and inconsistent descriptions, I did not mark the suggested revisions’ part, but made point-to-point revisions and detailed revisions records. As for the sentence revisions in the manuscript due to misspellings, grammar errors, and inconsistent descriptions, I have highlighted in red in the revised manuscript. We hope that the revision is acceptable, and I look forward to hearing from you soon. Additionally, the detailed correction records of error that you pointed out are as follows: 

Point 1: Simple Summary:"related resistance genes were screened and their action mechanism was discussed through the differential expression of transcriptome after pathogen infection" - It would be better to say "the action mechanism of these resistance genes was studied by analyzing the differential expression of the transcriptome after pathogen infection."

"to provide theoretical references for resistance breeding of soybean" - It would be better to say "to provide theoretical references for the breeding of soybean with resistance to diseases."

"The result of this study is that four genes were identified and it was found that the mode of its participation and coordination is to regulate the concentration of inorganic ions inside and outside the membrane through the transmembrane ion channel, and intervene the body regulation in stress by signal transduction." - It would be better to say "The study identified four genes that regulate the concentration of inorganic ions inside and outside the membrane through the transmembrane ion channel and participate in the body's stress regulation through signal transduction."

Response: Thanks for teacher’s suggestion, I replaced them with the sentence you provided, one by one.

Point 2: Introduction

In the first sentence, "in all countries all over the world" is redundant and could be simplified to "worldwide."

In the third sentence, "poor quality of commercial soybeans" should be "poor quality commercial soybeans" (remove the preposition "of").

In the eighth sentence, "body" should be "bodily" for clarity.

Response: Thank you very much for your patient review and detailed suggestions. I corrected these problems according to your suggestions.

Point 3: MM
In the first sentence, "seven leguminous families" should be changed to "seven leguminous species" to accurately reflect the context.

Response: According to teacher’s suggestion, I changed "seven leguminous families" to "seven leguminous species".

Point 4: What is NG method for dispersion?

Response: Nei-Gojobori (NG) method computes the numbers of synonymous and nonsynonymous substitutions and the numbers of potentially synonymous and potentially nonsynonymous sites. The specific methods, please refer to Nei and Gojobori (1986) documents, and I added this reference after teacher's suggestion.

Point 5: Line 122: It’s not bacteria, its fungi. "Molecular identification of pathogenic fungi."

In the sentence "To clarify the possible regulatory mechanism of CNGC gene in the abiotic or biological stress response, we use PlantCARE," "use" should be changed to "used" for consistency in tense.

In the paragraph on the isolation, purification, and morphological identification of the pathogen, "PDA culture dish" should be "PDA agar plate" to accurately reflect the context.

In the sentence "Activate the fungi to be cultured = with PDA medium," "Activate" should be replaced with "Inoculate" to accurately reflect the context.

In the sentence "After colonies grew to 2/3 of the dish, bacterial cakes were crushed," "bacterial cakes" should be replaced with "fungal colonies" to accurately reflect the context.

Response: Thanks teacher again and thank you very much for your patient review. These errors were corrected one by one.

Point 6: Results

You need to describe the methods that were used to reconstruct the phylogeny.

Response: This section, the structure and the classification of Arabidopsis thaliana sequences were used as a reference, seven legumes (Glycine max, Medicago truncatula, Phaseolus vulgaris, Cajanus cajan (Pigeon Pea), Lotus japonicus, Vigna angularis, Cicer Arietinum) and Arabidopsis thaliana CNGC genes are compared. And then the seven legumes are identified and classified according to the comparison results. And the detailed method of data obtained and phylogeny tree constructed was showed in “Materials and Methods”.

Point 7: Question: In the sentence "35, 17, 11, 19, 22, 21 and 18 CNGC genes containing two basic domains (PF00520/PF07885 and PF00027) and one CNGC-specific motif were identified, respectively," the word "respectively" is used unclear.

Response: After getting teacher’s suggestion, I deleted this word "respectively" after my careful consideration.

Point 8: In the sentence "Among the seven legumes, the proportion of members in group III was the largest and that in group II was the smallest," the word "that" is missing an antecedent.

Response: I changed “that” to “the proportion of members”.

Point 9: In the sentence "The Ka/Ks ratio can be used to measure the evolutionary selection of coding sequences," "evolutionary selection" is not a commonly used term and would be better phrased as "evolutionary pressure."

Response: Thanks teacher’s suggestion, I changed "evolutionary selection" to “"evolutionary pressure".

 Point 10: In the sentence "Interestingly, our prediction of AtCNGC19 and AtCNGC20 was located on the chloroplast membrane, but previous studies have verified that AtCNGC19 and AtCNGC20 in Arabidopsis are not confined to chloroplasts," the phrase "was located" should be "were located" to agree with the plural subject "AtCNGC19 and AtCNGC20."

Response: I feel sorry, this error was corrected, I changed the phrase "was located" to "were located".

 Point 11: In the sentence "The discovery of these elements provided a possibility for our subsequent experiments," "our subsequent experiments" is unclear and could be more specific

Response: After gaining teacher’s suggestion, I rewrote this sentence “The discovery of these elements provided a possibility for our subsequent experiments, which meant we should do some experiment for the functional verification of CNGCs” to explain the "our subsequent experiments".

Reviewer 4 Report

1.       Majority of the cases (especially in the methodology and result section) the proper pattern for the scientific names of different crops are not followed. Sometimes, the first letter in the species name is capitalized.

2.       For isolation of the pathogen, after treatment with sodium hypochlorite (NaOCl) 75% ethanol was used. I want to know about the utility of 75% ethanol and its scientific references.

3.       During isolation and pathogenicity detection of pathogens, multiple times authors write about ‘bacteria’. Are the authors confused about the isolation and pathogenicity proof process? Or they used a protocol prescribed for plant-pathogenic bacteria?

4.       Provide the accession number of the pathogen culture.

5.       Which method is followed for the differential gene expression…Transcriptome assay or qPCR analysis?

6.       Kindly provide the raw data of the transcriptome in the supplementary file.

Author Response

Thank you very much for your letter and advice. We have revised the paper, and would like to resubmit it for your consideration. In order to distinguish the revisions after getting your suggestions and the revisions in the manuscript due to misspellings, grammar errors, and inconsistent descriptions, I did not mark the suggested revisions’ part, but made point-to-point revisions and detailed revisions records. As for the sentence revisions in the manuscript due to misspellings, grammar errors, and inconsistent descriptions, I have highlighted in red in the revised manuscript. And the detailed correction records of error that you pointed out are as follows: 

Point 1: Majority of the cases (especially in the methodology and result section) the proper pattern for the scientific names of different crops are not followed. Sometimes, the first letter in the species name is capitalized.

Response: As for this question, I proofread the whole article and tried my best to avoid the above errors. 

Point 2: For isolation of the pathogen, after treatment with sodium hypochlorite (NaOCl) 75% ethanol was used. I want to know about the utility of 75% ethanol and its scientific references.

Response: Please listen to my explanation: in this study, the role of alcohol is to rinse and ensure no pollution by other bacterium of fungus in the process, and the role of sodium hypochlorite (NaOCl) is to kill bacterium of fungus under the samples.

Its early scientific reference is the book of Fang ZD, which provides some methods for plant diseases research.

Point 3: During isolation and pathogenicity detection of pathogens, multiple times authors write about ‘bacteria’. Are the authors confused about the isolation and pathogenicity proof process? Or they used a protocol prescribed for plant-pathogenic bacteria?

Response: I feel very sorry, this my default and I try to make up for the mistake.

Point 4: Provide the accession number of the pathogen culture.

Response: The accession number of the pathogen culture is OQ519857. After teacher’s suggestion, I uploaded this sequence.

Point 5: Which method is followed for the differential gene expression…Transcriptome assay or qPCR analysis?

Response: The method for the differential gene expression was transcriptome assay.

Point 6: Kindly provide the raw data of the transcriptome in the supplementary file.

Response: Due to the raw data of the transcriptome was too large, I had submitted to SRA database with the following project numbers: PRJNA849746, PRJNA849746, PRJNA849746, PRJNA849746, PRJNA849746, PRJNA849746, PRJNA849746, PRJNA849746, PRJNA849746, PRJNA849746, PRJNA849746, PRJNA849746, PRJNA849746, PRJNA849746, PRJNA849746, PRJNA849746, PRJNA849746, PRJNA849746, this point was showed in “Data Availability Statement”.

Round 2

Reviewer 1 Report

Almost all the concerns that I raised in my first review have been tackled. There is still one thing, which is that the manuscript still have too many figures. I still suggest the authors to reorganize their figures and some figures can be integrated into one.

Author Response

Question 1: Almost all the concerns that I raised in my first review have been tackled. There is still one thing, which is that the manuscript still have too many figures. I still suggest the authors to reorganize their figures and some figures can be integrated into one.

Response: Thank you again for your this suggestion. And for this suggestion, I integrated the original figure 6, figure 7 and figure 8 after careful consideration, and I think other figures had different meanings so that their cannot be integrated into one. I hope my response could satisfy you.

Reviewer 4 Report

The following information is missing

1. Authors, please provide the details of pathogen culture sequence information, also I could not find the details of the sequences in the NCBI which you provided the genebank accession number. 

2. Please provide RAW data weblink or online information which I can access it.

Author Response

Question 1: Authors, please provide the details of pathogen culture sequence information, also I could not find the details of the sequences in the NCBI which you provided the genebank accession number.

Question 2: Please provide RAW data weblink or online information which I can access it.

Response: We have carefully checked these two problems and ensured that the RAW data has been uploaded on NCBI, and the weblink is as follows:

1:https://www.ncbi.nlm.nih.gov/nuccore/OQ519857.1/

2:https://www.ncbi.nlm.nih.gov/sra?linkname=bioproject_sra_all&from_uid=849746 → https://trace.ncbi.nlm.nih.gov/Traces/?view=run_browser&acc=SRR19763942&display=metadata

I express my sincere thanks to your patient review and hope my response can satisfy you.